**Data Availability Statement:** This article presents all the raw data utilized in this study.

**Funding:** The author(s) received no specific funding for this work.

# Knowledge and perceptions of snakes, snakebites and their management among health care workers in Sudan

Ali Awadallah Saeed[1,2]*, Omer A. Gibreel[3], Ayman B. Mousa[1], Saeed M. Omer[4], Abdallateif Alkhair Omer[5], Intisar A. M. A. Elalawy[6], Ahmed Hassan Fahal[2]

1 Department of Pharmacology and Therapeutics, Faculty of Clinical and Industrial Pharmacy, National University-Sudan, Khartoum, Sudan, 2 Mycetoma Research Center, University of Khartoum, Khartoum, Sudan, 3 Accounting & Management Information Systems Department, College of Business Administration, Gulf University for Science & Technology, Mubarak Al-Abdullah, Kuwait, 4 Faculty of Medicine and Health Sciences, Gadarif University, Al Qadarif, Sudan, 5 Faculty of Medicine, Omdurman Islamic University, Omdurman, Sudan, 6 Alrayan College of Health Sciences and Nursing, Pharmacy Program, Almadina, Saudi Arabia

* aliawadsaeed@nu.edu.sd

## Abstract

### Background

Snakebite statistics in Sudan are lacking despite the high estimated burden of the problem. One study in Sudan reported the presence of 17 medically significant snakes belonging to three major families: Burrowing asps, Elapidae, and Viperidae. These snakes usually become abundant during and after the rainy season, and most snakebite victims are farm workers. This study was set out based on the observed snakebite management, poor outcomes and lack of information on the healthcare provider's knowledge of this serious, deadly medical and health condition in snakebite-endemic regions of Sudan.

### Materials and methods

In August 2022, a descriptive cross-sectional survey was conducted involving 394 medical and healthcare providers in snakebite-endemic regions of Sudan (Gaddarif, Sinnar, Khartoum, and Kassala). A validated questionnaire was used. It consisted of seven sections addressing the study population demographic characteristics, knowledge of snakes, snakebites, and their management. Data analysis used various statistical tests using Microsoft Excel and the Statistical Package for Social Sciences (SPSS) version 20 (IBM SPSS Inc., Chicago, IL) was done.

### Results

Among the 394 participants (44.7% males, 53.3% females), 58.1% demonstrated adequate knowledge of snakes, and 45.3% exhibited adequate knowledge of snakebites. A mere 25.9% received training in snakebite management, with 60.4% possessing adequate knowledge in this domain. Only 14% expressed high confidence in managing snakebites, and 40.9% reported having protocols for snakebite management at their health facilities.

**Competing interests:** The authors have declared that no competing interests exist.

## Conclusion

The study highlighted the inadequacy of healthcare providers' knowledge in snakebite-endemic areas in Sudan regarding snakes, snakebites and snakebites management. Urgent interventions, such as intensive continuing professional education and training, are essential to address this neglected medical and health problem.

## Introduction

Snakebite is considered one of the world's most serious medical problems. Snakebite was a neglected disease in the past, the World Health Organiztion (WHO) declared snakebite envenomation a neglected tropical disease On June 9[th], 2017 [1]. It called for new strategies to combat it and resources to implement them [2]. There are 5.4 million venomous snake bites globally, with 1.8 to 2.7 million cases of envenomings. [3]. Around 81,410 to 137,880 people die each year because of snake bites, and around three times as many amputations and other permanent disabilities are caused by snakebites annually. [1]. WHO reported that 81–95% of snake bites occur in the tropical regions of South Asia, Southeast Asia, Sub-Saharan Africa and Latin America [4]. In sub-Saharan Africa, the number of persons treated in health centers for snakebite envenomation is estimated at 315,000 cases per year, with more than 9,000 amputations and 7,000 deaths. Since many victims do not seek medical care, they are invisible to official reports; thus, the number of snake envenomation remains underestimated [5].

In some parts of the African continent, the snakebites' burden is higher than that of trypanosomiasis, leishmaniasis, and onchocerciasis, and its mortality can be higher than malaria [6]. The snakebites' high mortality in Africa is multifactorial, including lack of effective antitoxins, inadequate health care system, and scarcity of rapid access to medical centers [7]. Furthermore, many inhabitants of rural areas believe in traditional healers and do not seek treatment due to low socioeconomic and health education levels, many victims die shortly after the snakebite accident, and that is not reported in the hospital records [8]. In rural regions, where most snakebites occur, the general population's awareness of appropriate behaviour in response to snakebites is poor [9]. A study in Nigeria showed that most patients had attempted at least one potentially deleterious first-aid measure prior to presentation to hospitals [10]. In Ghana, an intervention to improve snakebite medical care ability and adherence to protocols significantly improved care and decreased snakebite mortality [11].

Snakebite statistics in Sudan are lacking despite the high estimated burden of the problem [12]. During the last 70 years, only a few data have been reported on Sudanese venomous snakes, and the Natural History Museum, University of Khartoum, reported some [13, 14]. Based on previous study which reported the presence of 17 medically significant snakes belonging to three major families: Burrowing asps, Elapidae, and Viperidae [15, 16]. These snakes usually become abundant during and after the rainy season, and most snakebite victims are farm workers [17].

This study was set out based on the observed snakebite management, poor outcomes and lack of information on the healthcare provider's knowledge of this serious, deadly medical and health condition in snakebite-endemic regions of Sudan (Gaddarif, Sinnar, Khartoum, and Kassala) [14].

## Material and methods

This descriptive cross-sectional survey was conducted in 15 August 2022 to 20 September 2022 using snowball sampling. The study participants were from snakebites envenomation endemic regions of Sudan. It included 394 medical and healthcare providers. After a review of the literature [5, 8], the standardized questionnaire prepared in English, and the draft questionnaire validated by authors for review and comments in order to ensure applicability, suitability, and consistency in environment of Sudan, furthermore a pilot test was conducted to further support the face and content validity.

The Sudan Snakebite Research Network developed this questionnaire. The questionnaire was widely distributed online, and all participants gave written informed consent. The questionnaire comprised seven sections covering demographic characteristics, knowledge of snakes (eight questions), knowledge of snakebites (nine questions), practice in dealing with snakebites (five questions), attitude toward affected patients (two questions), and snakebite management (ten questions).

### Data analysis

The data underwent entry and management using Microsoft Excel and the Statistical Package for Social Sciences (SPSS) version 20 (IBM SPSS Inc., Chicago, IL). Frequency and percentage were computed for qualitative variables, while quantitative variables were characterised by median and interquartile range (IQR). A comprehensive score encompassing knowledge, attitude, and practice was determined and subsequently categorised as adequate or inadequate based on the average score. Percentages were employed to articulate attitudes towards dealing with snakebites. The analysis of questions involved scoring each correct answer as 1, with "don't know" and incorrect responses receiving 0 points. The result is expressed as percentages. The results were rated adequate and inadequate based on the median.

### Ethical considerations

Ethical clearance for the study was obtained from the National University Institutional Review Board numbered **NU-REC/11-018/12**, and all participants gave written informed consent.

## Results

The study included 394 medical and healthcare providers, with 176 (44.7%) males and 218 (53.3%) females. Among the participants, 142 (36%) were under 40, while 252 (64%) were 40 or older with average age (29.3 ± 8). The distribution of participants included pharmacists (31%), medical officers (23.9%), registrars (20%), house officers (11.7%), nurses (6.8%), and consultants (6.6%), Table 1.

Analysis revealed that 255 participants (64.7%) had less than five years of experience in medical and health practice, 89 (22.3%) had experience ranging from five to ten years, and 50 (13%) possessed more than ten years of experience. Regarding training in snakebite management, only 102 participants (25.9%) had received such training. Among them, 37% acquired it from senior colleagues, 28.5% were self-educated, and 10% obtained it through training courses, Table 1.

There were eight questions on snake knowledge, and the results showed that 321 (81.5%) knew the fact that not all snakes are poisonous, 129 (32.7%) stated all snakes are carnivorous, 255 (64.7%) knew not all snakes are venomous, 113 (28.7%) recognised all snakes have fangs in front of their mouth, 213 (54.1%) thought snakes pick sounds using their ears, 213 (54.1%) believed snakes are important for farmers, and 255 (64.7%) believed deforestation and

**Table 1. Socio-demographic characteristics of the study participants (n = 394).**

| Variables | Variable | No. (%) |
|---|---|---|
| Sex | Male | 176 (44.7) |
| | Female | 218 (53.3) |
| Age | 21–30 | 55 (14) |
| | 31–40 | 87 (22.1) |
| | 41–50 | 139 (35.3) |
| | Above 50 | 113 (28.6) |
| Occupation | Medical officer | 94 (23.9) |
| | House officer | 46 (11.7) |
| | Registrar | 79 (20) |
| | Consultant | 26 (6.6) |
| | Pharmacist | 122 (31) |
| | Nurse | 27 (6.8) |
| Working Sector | Governmental | 229 (58.1) |
| | Private | 165 (41.9) |
| Years of practice | < 5 years | 255 (64.7) |
| | 5–10 years | 89 (22.3) |
| | >10 years | 50 (13) |
| Attended training on snakebite management | Yes | 102 (25.9) |
| | No | 292 (74.1) |
| Skills in snake bite management | Self-education (internet or textbooks) | 109 (28.5) |
| | Learning from senior colleagues | 142 (37) |
| | Knowledge and skills gained during training | 39 (10.2) |
| | Don't Know | 94 (24.3) |

urbanisation have increased human-snake interaction, Table 2). The study showed that 322 participants (81.7%) knew snakes are reptiles, Table 3. Moreover, 229 participants (58.1%) answered snake knowledge questions correctly, while 165 (41.9%) had incorrect answers.

There were nine questions concerning snakebite knowledge, and 85 participants (21.6%) responded correctly to the statement that handling a dead snake's head is not safe enough. Additionally, 195 (49.5%) recognised the fact that marks can always be seen or found on the victim after every snakebite, and 219 (55.6%) believed that a person reported at the hospital with symptoms of snakebite could be given an antitoxin injection without actually being bitten by a snake. Moreover, 122 (31%) believed sleeping under mosquito nets can prevent snake-bites. A total of 151 (38.3%) did not think the statement that a venomous ("poisonous") snake always injects venom (poison) into the victim. Furthermore, 316 (80.2%) recognised that the snake type determines the symptoms and signs of snakebites, and 279 (70.8%) thought that the symptoms and signs of the snakebite depend on the amount of venom injected by the snake, Table 2. The obtained data showed that 209 participants (53%) believed the symptoms and signs could determine the snake and venom injected, and 31 (7.9%) felt the snakebite commonly occurs during the day, Table 3. For snakebite knowledge, 179 participants (45.3%) answered the questions correctly, while 215 (54.7%) responded incorrectly.

The study showed that 247 participants (26.7%) believed that a tourniquet prevents the spread of poison in the remaining part of the body, and 320 (81.2%) stated reassurance of the victim should be practiced as the first protocol for prevention. Regarding the belief that an incision at the bite site helps remove the poison, 186 participants (47.2%) held this belief, and 282 (71.6%) thought if the leg is the bite site, its elevation will not reduce poison spread.

**Table 2. The study participants knowledge of snakes and snakebites (n = 394).**

| | Yes (%) | No (%) | I don't know (%) |
|---|---|---|---|
| **Snake Knowledge** | | | |
| All snakes are venomous | 36 (9.1) | 321 (81.5) | 37 (9.4) |
| All snakes are carnivorous | 129 (32.7) | 156 (39.6) | 109 (27.7) |
| All snakes are venomous | 94 (23.9) | 255 (64.7) | 45 (11.4) |
| All snakes have fangs in front of their mouth | 113 (28.7) | 164 (41.6) | 117 (29.7) |
| Snakes pick sounds using their ears | 71 (18) | 173 (43.9) | 150 (38.1) |
| Snakes are important for farmers | 213 (54.1) | 89 (22.6) | 92 (23.3) |
| Deforestation and urbanisation have increased human-snake interaction | 255 (64.7) | 57 (14.5) | 82 (20.8) |
| **Snakebites Knowledge** | | | |
| Handling a dead snake's head is safe | 209 (53) | 85 (21.6) | 100 (25.4) |
| Marks can always be seen on the victim after every snake bite? | 161 (40.9) | 195 (49.5) | 38 (9.6) |
| Can a person report at the hospital with symptoms of snake bite without actually being bitten by a snake? | 219 (55.6) | 95 (24.1) | 80 (20.3) |
| Sleeping under mosquito nets can prevent snakebites | 122 (31) | 223 (56.6) | 49 (12.4) |
| A venomous snake bite always injects venom into the victim | 172 (53.7) | 151 (38.3) | 71 (18) |
| Signs and symptoms of snake bites are determined by the type of snake responsible for the bite. | 316 (80.2) | 32 (8.1) | 46 (11.7) |
| The signs and symptoms of a snakebite depend on the amount of venom injected by the snake. | 279 (70.8) | 60 (15.2) | 55 (14) |

Moreover, 297 participants (75.4%) believed bringing the snake to the treating physician increases the chances of survival by correctly identifying the snake, Table 4. In this study, 238 participants (60.4%) answered snakebite practice questions correctly, while 156 (39.6%) did not.

Concerning attitudes toward snakebites, residing in cities as a protective factor against snakebites was stated by 297 participants (75.4%), and 106 (26.9%) believed that snakebites result from revenge inspired by past incidents, Table 5.

**Table 3. The study participants knowledge of snakes and snakebites (n = 394).**

| Variable | Categories | Frequency (%) |
|---|---|---|
| **Snakebite Knowledge** | | |
| How to identify a snake bite | Type of snake | 78 (19.8) |
| | Signs and symptoms | 209 (53) |
| | Victim presenting complaints | 55 (14) |
| | I don't know | 52 (13.2) |
| Time of bite | During the day | 31 (7.9) |
| | During the night | 274 (69.5) |
| | Not sure | 65 (22.6) |

**Table 4. The study participants' practice regarding snakebites (n = 394).**

| Variable | Yes (%) | No. (%) |
|---|---|---|
| A tourniquet prevents the spread of venom in the remaining part of the body | 247 (26.7) | 147 (73.3) |
| Reassurance of the victim should be practiced as the 1st protocol for prevention | 320 (81.2) | 74 (18.8) |
| Incision at the site of bite helps removing venom | 186 (47.2) | 208 (52.8) |
| If the bite site is leg, elevating it will reduce the spread of venom | 112 (28.4) | 282 (71.6) |
| Bringing snake to treating physician increases chances of survival by correct identification | 297 (75.4) | 97 (24.6) |

The study showed that only 55 participants (14%) rated themselves as very confident in managing snakebites, and 203 participants (51.5%) had previously managed a snakebite patient. Regarding management and help provided by participants in snakebite management in 2021, only 20 participants (5.1%) managed more than 100 patients, and 56 (14.2%) managed between 10 and 50 patients. Concerning the availability of snakebite management protocol, 161 (40.9%) stated they have that at their health facilities. Only 179 participants (45.4%) said their health facility has what it takes to manage snakebites effectively, Table 6.

Regarding the management of snakebites, 265 participants (67.3%) considered it an emergency. Additionally, 178 (45.2%) usually admit snakebite patients to hospitals, while 26 participants (6.6%) usually refer patients immediately to other centers. (Table 7)

Sixty-one participants (15.5%) correctly said a 20-minute whole blood count test is the most appropriate test for snakebite management. The statement that antivenoms are the only antidotes in managing snakebites by venomous snakes was agreed upon by 267 participants (75.4%). (Table 7)

Moreover, 199 participants (50.5%) stated antivenoms made anywhere in the world are suitable for managing snakebite victims, and 191 (48.4%) thought that antivenoms should be given to all patients bitten by snakes. Fifty-seven participants (14.5%) believed that the intramuscular injection of an anti-snake venom is as effective as an intravenous one. Furthermore, 197 participants (50%) did not know how long the anti-snake venoms could remain useful after the expiry date. Regarding anti-snake venoms, 132 participants (33.6%) believed it is better to give low doses repeated over several days than high initial doses. (Table 7)

## Discussion

Snakebites represent a substantial global medical and health challenge, with the potential for fatal consequences, especially prominent in specific regions [12]. The repercussions extend beyond immediate health concerns, encompassing a significant economic impact on individuals and communities [18].Those affected by snakebites may encounter enduring disabilities, leading to the loss of livelihoods and heightened poverty, thereby sustaining a cycle of disadvantage [19]. In Sudan, most of the basic information on snakebites is lacking; likewise, the medical and healthcare workers knowledge of this fatal disease, so this study was set out. [14]

In this study, most participants (64.7%) possessed less than five years of practical experience. Only a quarter (25.9%) underwent snakebite management training, which was informal

**Table 5. The study participants attitude regarding snakebites (n = 394).**

| Variable | Yes (%) | No (%) |
|---|---|---|
| Residing in cities is a protective factor from snake bite | 297 (75.4) | 97 (24.6) |
| Snakebite is an outcome of revenge inspired by past incidents | 106 (26.9) | 288 (73.1) |

Table 6. Management of snakebites (n = 394).

| Variable | Response | No. (%) |
|---|---|---|
| Management of snake bites | Very confident | 55 (14) |
| | Confident | 106 (26.9) |
| | Fairly confident | 132 (33.5) |
| | Not confident | 101 (25.6) |
| Have you ever managed, supervised or nursed a snakebite patient before? | Yes | 203 (51.5) |
| | No | 191 (48.5) |
| Prevalence of snakebites seen by HCWs | One person every day | 08 (2) |
| | 1 to 5 persons a week | 76 (19.3) |
| | More than five persons a week | 22 (5.6) |
| | 1 to 5 persons every month | 46 (11.7) |
| | More than five persons every month | 06 (1.5) |
| | One person every three months | 11 (2.8) |
| | More than five persons every three months | 07 (1.8) |
| | 1 to 5 persons in the year | 28 (7.1) |
| | More than five persons in the year | 05 (1.3) |
| | Not Sure | 185 (46.9) |
| The health facility has a protocol for the management of snakebites | Yes | 161 (40.9) |
| | No | 144 (36.5) |
| | I don't know | 89 (22.6) |
| The health facility has what it takes to effectively manage snake bites | Yes | 179 (45.4) |
| | No | 215 (54.6) |

in 90% of them. These findings are serious as the study participants practicing in snakebites endemic regions. These findings align with a report from Kenya Uganda, and Zambia, which showed only 12% of the healthcare workers received formal snakebite management training [20]. These findings necessitate designing objectives and continuing professional development training and education for healthcare providers, particularly those working in snakebite-endemic regions of Sudan (Gaddarif, Sinnar, Khartoum, and Kassala) [14].

Based on previous studies, managing snakebites affected patients can be challenging, particularly in regions where venomous snakes are prevalent, and healthcare resources may be limited [21]. The difficulties and constraints associated with snakebite management include the limited access to healthcare, lack of antivenom availability, variability in snake venom, difficulty in snake identification, inadequate training of healthcare personnel, financial constraints, cultural beliefs and practices, lack of public awareness, insufficient research and data and climate and environmental factors [22]. These challenges require a comprehensive approach involving improved healthcare infrastructure, better access to antivenom, education and training for healthcare professionals and communities, and increased research to enhance our understanding of snakebite dynamics [21]. All these should be addressed in managing these patients, designing continuing professional development activities and prevention programmes. International collaboration and support are also crucial to developing effective strategies for snakebite management in diverse regions [23].

As stated in table [7] in this study, the participants' knowledge of antivenom was inadequate to lack of knowledge in answering antivenom questions. In Sudan, numerous endemic areas lack access to antivenom, resulting in severe consequences. Multiple factors contribute to the constrained availability of quality antivenom in regions affected by snakebites [24]. The

**Table 7. The study participants knowledge of snakebites management (n = 394).**

| Statement | Response | No. (%) |
|---|---|---|
| How do you triage (prioritise) someone with a snakebite in your facility? | Emergency | 265 (67.3) |
| | Urgent | 102 (25.9) |
| | Not urgent | 07 (1.8) |
| | I don't know | 20 (5) |
| What do you do when people report to your health facility with snakebite? | Refer immediately | 26 (6.6) |
| | Give first aid treatments and refer | 166 (42.1) |
| | Admit and treat | 178 (45.2) |
| | Call for assistance from another health facility | 24 (6.1) |
| Which of the following tests will you first recommend when someone reports snakebite to determine there was actually an injection of venom into the person (envenoming)? | Full blood count | 182 (46.2) |
| | Grouping and cross matching | 65 (16.5) |
| | 20 minutes whole blood count test | 61 (15.5) |
| | Urinalysis for myoglobinuria | 86 (21.8) |
| Antivenoms are the only antidotes in managing snake bites by venomous snakes. | Yes | 267 (75.4) |
| | No | 58 (7.1) |
| | I don't know | 69 (17.5) |
| Antivenoms made anywhere in the world is good for the management of snake bite victims | Yes | 199 (50.5) |
| | No | 83 (21.1) |
| | I don't know | 112 (28.4) |
| Should antivenoms be given to all patients bitten by snakes? | Yes | 191 (48.4) |
| | No | 161 (40.9) |
| | I don't know | 42 (10.7) |
| When an anti-snake venom is injected into the muscle (intramuscular), it is as effective as when injected into the veins (intravenous) | Yes | 57 (14.5) |
| | No | 205 (52) |
| | I don't know | 132 (33.5) |
| Anti-snake venoms remain useful for months or even years after stated expiry dates | Yes | 45 (11.4) |
| | No | 152 (38.6) |
| | I don't know | 197 (50) |
| In using anti-snake venoms, it is better to give low doses repeated over several days than high initial doses. | Yes | 132 (33.6) |
| | No | 118 (29.9) |
| | I don't know | 144 (36.5) |

antivenom high production cost renders it unaffordable for many patients. Additionally, numerous distribution challenges stemming from limited infrastructure, poor transportation networks, and inadequate storage facilities further impede antivenom's efficient and timely dissemination to affected areas. [10] Complicating matters is the diverse array of snake species and the variability in venom composition, necessitating the production of different antivenoms tailored to specific snakebites, which is impossible in the country [25]. Medical and healthcare providers should be aware of these issues.

Based on previous study, in many parts of the country, cultural values and traditional healing procedures may guide the decision-making process for snakebite victims and families. Some individuals may prefer traditional remedies over modern medical care, resulting in delayed treatment. Hence, healthcare providers need to organise objective health education sessions and events to reduce the burden of late presentation with massive disease and complications [26].

It is clear from this study that healthcare providers lack snakebite-effective management, and this is in line with many reports from the African continent. [27, 28]. Thus, structured and objective training and educational programmes should provided within the medical and health undergraduate and postgraduate curricula. The data revealed that only 40.9% of our participants have snakebite management protocols in their health facilities, which is low in accordance with report from Ghana (73.7%) [29], which is one of challenges in Uganda (lack of up-to-date robust clinical guidelines) [17]. Snakebite management guidelines are crucial tools that improve healthcare interventions' consistency, quality, and effectiveness. These guidelines significantly reduce the morbidity and mortality associated with snakebites by providing a structured framework for assessment, treatment, and prevention.

Overall, the participants in the study exhibit insufficient knowledge regarding snakes and the management of snakebites, and there is a notable absence of reports on these matters. This underscores the need to conduct surveillance studies to understand their epidemiology better. Such studies are essential for formulating targeted control and prevention measures based on objective insights [9].

As with other neglected tropical diseases, [30] training healthcare providers on snakebite management using digital facilities is important in improving patient outcomes. It not only addresses accessibility and cost considerations but also promotes interactive, consistent, and up-to-date learning. Such training strategies enhance the preparedness and effectiveness of healthcare professionals in managing snakebite cases.

The primary strength of the present study lies in its substantial sample size, comprising healthcare providers operating in endemic regions. This sample offers valuable insights into the knowledge base of practitioners involved in treating patients affected by snakebites. Nonetheless, conducting more comprehensive investigations is essential to discern details about local health facilities, management guidelines and protocols, patients presentations, and treatment outcomes. Such in-depth studies are crucial for sharing experiences and understanding the intricacies associated with snakebite management in specific localities.

## Supporting information

**S1 File.**
(DOCX)

## Author Contributions

**Conceptualization:** Ali Awadallah Saeed, Saeed M. Omer, Intisar A. M. A. Elalawy, Ahmed Hassan Fahal.

**Data curation:** Ayman B. Mousa, Saeed M. Omer, Abdallateif Alkhair Omer.

**Formal analysis:** Omer A. Gibreel.

**Methodology:** Ali Awadallah Saeed, Omer A. Gibreel, Ahmed Hassan Fahal.

**Supervision:** Ahmed Hassan Fahal.

**Validation:** Ali Awadallah Saeed, Omer A. Gibreel, Ahmed Hassan Fahal.

**Writing – original draft:** Ali Awadallah Saeed, Ayman B. Mousa, Abdallateif Alkhair Omer, Ahmed Hassan Fahal.

**Writing – review & editing:** Ayman B. Mousa, Saeed M. Omer, Abdallateif Alkhair Omer, Intisar A. M. A. Elalawy, Ahmed Hassan Fahal.

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
