## [Decision Letter · Decision Letter 0]

10 Nov 2023

PONE-D-23-32936Knowledge, Attitude, and Practices of Snakes, Snakebites and their Management among Health Workers in Sudan: Need for Internet-mediated Knowledge Management System for Continuing Professional Development.PLOS ONE

Dear Dr. Saeed,

Thank you for submitting your manuscript to PLOS ONE. After careful consideration, we feel that it has merit but does not fully meet PLOS ONE’s publication criteria as it currently stands. Therefore, we invite you to submit a revised version of the manuscript that addresses the points raised during the review process.

ACADEMIC EDITOR:The title could be revised to precisely read: **Knowledge and perceptions of snakes, snakebites and their management among health workers in Sudan**Please indicate how the sample size was arrived at.Provide the questionnaire used in the survey (possibly as a supplementary file)Avoid starting sentences with numbers.References need to be captured correctly using the same style.There are several studies with Africa which gives the perceptions about snakes and snakebites from both traditional healers and health workers. These could help the authors discuss their results meaningfully.  See for example;https://doi.org/10.1371/journal.pone.0291032https://doi.org/10.1016/j.toxcx.2021.100078https://doi.org/10.4269%2Fajtmh.20-1078https://doi.org/10.1186/s41182-019-0187-0https://doi.org/10.1093/inthealth/ihac043https://www.medrxiv.org/content/medrxiv/early/2023/02/23/2023.02.16.23286015.full.pdfThis submission could also benefit from English language proofreading by a proficient speaker.The rest of the suggestions are in the attached manuscript file.

We look forward to receiving your revised manuscript.

Kind regards,

Timothy Omara, PhD

Academic Editor

PLOS ONE

Journal Requirements:

Reviewers' comments:

Reviewer's Responses to Questions

**Comments to the Author**

1. Is the manuscript technically sound, and do the data support the conclusions?

Reviewer #1: Yes

Reviewer #2: No

2. Has the statistical analysis been performed appropriately and rigorously? 

Reviewer #1: Yes

Reviewer #2: Yes

3. Have the authors made all data underlying the findings in their manuscript fully available?

Reviewer #1: Yes

Reviewer #2: No

4. Is the manuscript presented in an intelligible fashion and written in standard English?

Reviewer #1: No

Reviewer #2: No

5. Review Comments to the Author

Reviewer #1: A well-conducted survey showed the need to train health professionals to deal with patients bitten by venomous snakes.

Observe the correct use of the terms poisonous, venomous, venom and venom.

Poison is a toxin that gets into the body by inhaling, swallowing, or absorption through the skin (Poisonous animals = toads and pupper fish). Venomous animals (some snakes and scorpions): it's when the toxin is injected into you.

See attached the manuscript with a few corrections.

Reviewer #2: Terming snakes as venomous rather than calling it poisonous would be correct.

Title is not clear. I could not understand "Practices of Snakes", as there is no information of snake's practice in the manuscript. The conclusion is divorced from the result.

As manuscript contains rare and amazing information about the poor farmers problem and knowledge of health workers in this regard please connect conclusion to the findings. Manuscript is not quite in flow with Title-Result-Discussion and Recommendation.

Yes, recommendation seem fitting at abstract and manuscript but it would be misleading in the title itself.

Provide the data availability statement.

Repetition of result in discussion section and regular citation of same literature multiple times makes the discussion section of manuscript quite dull. Addition of multiple references from different part of the world with inference of the result would help in making better manuscript.

6. PLOS authors have the option to publish the peer review history of their article (what does this mean?). If published, this will include your full peer review and any attached files.

Reviewer #1: **Yes: **Paulo Sérgio Bernarde

Reviewer #2: **Yes: **Sunil Sapkota

---

## [Author Response · Author response to Decision Letter 0]

21 Nov 2023

1. The title was changed following your recommendation

2. We explain how the sample size was arrived at.

3. The survey will be provided.

4. Numbers were avoided in the starting of sentences. 

5. The references were formatted correctly using the same style.

6. Four studies from other African countries were added.

7. The submission benefit from English language proofreading by a proficient speaker.

8. The suggestions were in the attached manuscript file were included.

---

## [Decision Letter · Decision Letter 1]

22 Jan 2024

PONE-D-23-32936R1Knowledge and perceptions of snakes, snakebites and their management among health workers in Sudan.PLOS ONE

Dear Dr. Saeed,

Thank you for submitting your manuscript to PLOS ONE. After careful consideration, we feel that it has merit but does not fully meet PLOS ONE’s publication criteria as it currently stands. Therefore, we invite you to submit a revised version of the manuscript that addresses the points raised during the review process.

 Please submit your revised manuscript by Mar 07 2024 11:59PM. If you will need more time than this to complete your revisions, please reply to this message or contact the journal office at plosone@plos.org. Please include the following items when submitting your revised manuscript:A rebuttal letter that responds to each point raised by the academic editor and reviewer(s). You should upload this letter as a separate file labeled 'Response to Reviewers'.A marked-up copy of your manuscript that highlights changes made to the original version. You should upload this as a separate file labeled 'Revised Manuscript with Track Changes'.An unmarked version of your revised paper without tracked changes. You should upload this as a separate file labeled 'Manuscript'.

We look forward to receiving your revised manuscript.

Kind regards,

Timothy Omara, PhD

Academic Editor

PLOS ONE

Reviewers' comments:

Reviewer's Responses to Questions

**Comments to the Author**

1. If the authors have adequately addressed your comments raised in a previous round of review and you feel that this manuscript is now acceptable for publication, you may indicate that here to bypass the “Comments to the Author” section, enter your conflict of interest statement in the “Confidential to Editor” section, and submit your "Accept" recommendation.

Reviewer #2: All comments have been addressed

Reviewer #3: (No Response)

2. Is the manuscript technically sound, and do the data support the conclusions?

Reviewer #2: Partly

Reviewer #3: No

3. Has the statistical analysis been performed appropriately and rigorously? 

Reviewer #2: No

Reviewer #3: No

4. Have the authors made all data underlying the findings in their manuscript fully available?

Reviewer #2: No

Reviewer #3: No

5. Is the manuscript presented in an intelligible fashion and written in standard English?

Reviewer #2: Yes

Reviewer #3: No

6. Review Comments to the Author

Reviewer #2: Please be specific about which group of health workers lack knowledge on snake bite management. Mention the location of study area, mention in which health institution the study was conducted. Mention ratio of health workers to the ratio of health workers participated in the study.

Reviewer #3: Title: "Assessment of Knowledge and Practices among Medical and Healthcare Providers in Snakebite-Endemic Regions of Sudan"

The paper by Saeed et al is a report on the Knowledge and Practices among medical and healthcare providers i nsnakebite-endemic regions of Sudan'. Please see below my comments for your consideration. Further comments are provided i nthe attached document.

Abstract:

The manuscript under consideration purports to assess the knowledge and practices of medical and healthcare providers regarding snakebites in Sudan, a region grappling with the significant burden of this neglected tropical disease. While the topic is undoubtedly pertinent, the manuscript exhibits several critical flaws in methodology, analysis, and overall scientific rigor, casting serious doubts on its validity and contribution to the scientific literature.

Major Concerns:

1. Methodological Shortcomings:

- The manuscript's methodology lacks clarity, particularly in its description of the sampling technique. The use of snowball sampling raises questions about the representativeness of the study population. Additionally, the reliance on a self-administered web-based questionnaire introduces a selection bias, excluding individuals with limited access to the internet or those who may be less inclined to participate in online surveys.

- The questionnaire's validation process is inadequately reported, leaving the reader uncertain about the tool's reliability and validity. The lack of information regarding the pilot study's outcomes or adjustments made to the questionnaire undermines the study's scientific rigor.

2. Data Analysis and Reporting Issues:

- The data analysis section is notably deficient in critical details. The statistical tests employed are vaguely described, and the specific analyses performed using Microsoft Excel and SPSS lack transparency. A more detailed explanation of statistical methods is imperative for readers to assess the validity of the study's findings.

- The presentation of results lacks clarity and coherence. Essential information, such as confidence intervals and measures of variability, is conspicuously absent, hampering the reader's ability to interpret the significance and reliability of the reported results.

3. Incomplete Background and Literature Review:

- The introduction fails to provide a comprehensive background on the existing knowledge and research gaps in the field of snakebite management in Sudan.

4. Generalization and External Validity:

- The study claims to draw conclusions about the knowledge and practices of healthcare providers in snakebite-endemic regions of Sudan. However, the absence of a clear geographic delineation of the study areas and the use of a convenience sampling method limit the generalizability of the findings to the broader target population.

5. Ethical Considerations and Informed Consent:

- The manuscript lacks transparency regarding ethical considerations. While it mentions obtaining ethical clearance, critical details such as the reference number, and participant confidentiality measures are missing. This omission raises ethical concerns and undermines the manuscript's scientific integrity.

Conclusion:

In light of the aforementioned critical concerns regarding methodology, data analysis, literature review, generalization, and ethical considerations, it is my firm recommendation that this manuscript be rejected for publication. The scientific community deserves robust and rigorously conducted research, and this manuscript falls significantly short of meeting those standards. Addressing the outlined deficiencies is imperative for any future resubmission to be considered for publication.

7. PLOS authors have the option to publish the peer review history of their article (what does this mean?). If published, this will include your full peer review and any attached files.

Reviewer #2: **Yes: **Sunil Sapkota

Reviewer #3: **Yes: **Mitchel Okumu

---

## [Author Response · Author response to Decision Letter 1]

7 Mar 2024

Comments to reviewer:

Reviewer #3: Title: "Assessment of Knowledge and Practices among Medical and Healthcare Providers in Snakebite-Endemic Regions of Sudan"

The paper by Saeed et al is a report on the Knowledge and Practices among medical and healthcare providers in snakebite-endemic regions of Sudan'. Please see below my comments for your consideration. Further comments are provided in the attached document.

1. Methodological Shortcomings:

- The manuscript's methodology lacks clarity, particularly in its description of the sampling technique. The use of snowball sampling raises questions about the representativeness of the study population. Additionally, the reliance on a self-administered web-based questionnaire introduces a selection bias, excluding individuals with limited access to the internet or those who may be less inclined to participate in online surveys.

Author comment:

The used of snowball sampling has been extensively used in studies where hard to reach population problem was present, in our case we tried our best to collect samples of our respondents from 4 areas in Sudan known for the high rate of snake bite. We used the method stated by Raifman et al to sample hard to reach population. 

In RDS and time-location sampling (TLS), factors that influence inclusion can be estimated and accounted for in an effort to generate representative samples. RDS is particularly equipped to reach the most hidden members of hard-to-reach populations.

(Raifman, S., DeVost, M.A., Digitale, J.C. et al. Respondent-Driven Sampling: a Sampling Method for Hard-to-Reach Populations and Beyond. Curr Epidemiol Rep 9, 38–47 (2022). https://doi.org/10.1007/s40471-022-00287-8)

- The questionnaire's validation process is inadequately reported, leaving the reader uncertain about the tool's reliability and validity. The lack of information regarding the pilot study's outcomes or adjustments made to the questionnaire undermines the study's scientific rigor.

Author Comment:

After a review of the literature (8, 31), the standardized questionnaire prepared in English, and the draft questionnaire validated by authors for review and comments in order to ensure applicability, suitability, and consistency in our environment, as well as to define the face and content validity.

2. Data Analysis and Reporting Issues:

- The data analysis section is notably deficient in critical details. The statistical tests employed are vaguely described, and the specific analyses performed using Microsoft Excel and SPSS lack transparency. A more detailed explanation of statistical methods is imperative for readers to assess the validity of the study's findings.

- The presentation of results lacks clarity and coherence. Essential information, such as confidence intervals and measures of variability, is conspicuously absent, hampering the reader's ability to interpret the significance and reliability of the reported results.

Author comment:

Due to the nature of the study as descriptive exploratory study, none of the inferential statistical method was employed. This is line with numerous similar papers that publish such types of exploratory studies in area of snakes, snakebites and snakebite management. 

3. Incomplete Background and Literature Review:

- The introduction fails to provide a comprehensive background on the existing knowledge and research gaps in the field of snakebite management in Sudan.

Author comment:

In Sudan, snakebite statistics are lacking despite the high estimated burden of the problem [12]. During the last 70 years, only a few data have been reported on Sudanese venomous snakes, and the Natural History Museum, University of Khartoum, reported some [13,14]. One study reported the presence of 17 medically significant snakes belonging to three major families: Burrowing asps, Elapidae, and Viperidae [15]. These snakes usually become abundant during and after the rainy season, and most snakebite victims are farm workers [16]. 

4. Generalization and External Validity:

- The study claims to draw conclusions about the knowledge and practices of healthcare providers in snakebite-endemic regions of Sudan. However, the absence of a clear geographic delineation of the study areas and the use of a convenience sampling method limit the generalizability of the findings to the broader target population

Author Comment:

The study participants were from snakebites envenomation endemic regions of Sudan (Gaddarif, Sinnar, Khartoum, and Kassala) (figure 1)

5. Ethical Considerations and Informed Consent:

- The manuscript lacks transparency regarding ethical considerations. While it mentions obtaining ethical clearance, critical details such as the reference number and participant confidentiality measures are missing. This omission raises ethical concerns and undermines the manuscript's scientific integrity.

Author comment:

• Reference number: NU-REC/11-018/12

• Our study observed all the rights of the participants and conscious of the confidentiality of the information obtained and kept the anonymity of the participants during and after the scheduled period of the study. 

Reviewer #2: Please be specific about which group of health workers lack knowledge on snake bite management. Mention the location of study area, mention in which health institution the study was conducted. Mention ratio of health workers to the ratio of health workers participated in the study.

Author comment:

In table 1 the healthcare workers participated in the study were there. 

Ali Awadallah Saeed 

Corresponding author

---

## [Decision Letter · Decision Letter 2]

24 Mar 2024

PONE-D-23-32936R2Knowledge and perceptions of snakes, snakebites and their management among health workers in Sudan.PLOS ONE

Dear Dr. Saeed,

Thank you for submitting your manuscript to PLOS ONE. After careful consideration, we feel that it has merit but does not fully meet PLOS ONE’s publication criteria as it currently stands. Therefore, we invite you to submit a revised version of the manuscript that addresses the points raised during the review process. Please submit your revised manuscript by May 08 2024 11:59PM. If you will need more time than this to complete your revisions, please reply to this message or contact the journal office at plosone@plos.org. Please include the following items when submitting your revised manuscript:A rebuttal letter that responds to each point raised by the academic editor and reviewer(s). You should upload this letter as a separate file labeled 'Response to Reviewers'.A marked-up copy of your manuscript that highlights changes made to the original version. You should upload this as a separate file labeled 'Revised Manuscript with Track Changes'.An unmarked version of your revised paper without tracked changes. You should upload this as a separate file labeled 'Manuscript'.If applicable, we recommend that you deposit your laboratory protocols in protocols.io to enhance the reproducibility of your results. Protocols.io assigns your protocol its own identifier (DOI) so that it can be cited independently in the future. For instructions see: https://journals.plos.org/plosone/s/submission-guidelines#loc-laboratory-protocols. Additionally, PLOS ONE offers an option for publishing peer-reviewed Lab Protocol articles, which describe protocols hosted on protocols.io. Read more information on sharing protocols at https://plos.org/protocols?utm_medium=editorial-email&utm_source=authorletters&utm_campaign=protocols.

We look forward to receiving your revised manuscript.

Kind regards,

Timothy Omara, PhD

Academic Editor

PLOS ONE

Journal Requirements:

Additional Editor Comments:

Dear authors,

In addition to the reviewer comments, please revise your manuscript taking into considering some minor concerns I identified with the current draft:

-In Table 1, the average age (29.3 ± 8) should be deleted, it should only be indicated in the main text. Please also provide the ages in some meaningful ranges. The current groups (< 40 and ≥ 40) does not give a good picture of the participants in this study.

-Table 3, the first aspect (Type of snakes) is not necessary, please remove it.

Reviewers' comments:

Reviewer's Responses to Questions

**Comments to the Author**

1. If the authors have adequately addressed your comments raised in a previous round of review and you feel that this manuscript is now acceptable for publication, you may indicate that here to bypass the “Comments to the Author” section, enter your conflict of interest statement in the “Confidential to Editor” section, and submit your "Accept" recommendation.

Reviewer #2: (No Response)

2. Is the manuscript technically sound, and do the data support the conclusions?

Reviewer #2: Partly

3. Has the statistical analysis been performed appropriately and rigorously? 

Reviewer #2: N/A

4. Have the authors made all data underlying the findings in their manuscript fully available?

Reviewer #2: Yes

5. Is the manuscript presented in an intelligible fashion and written in standard English?

Reviewer #2: Yes

6. Review Comments to the Author

Reviewer #2: There are some typo error in manuscript, specially in abstract section, please make proper edits. There are copy paste repetition of some lines from discussions in the abstract. Please be consistent with use of fonts and citation rules of the journal.

7. PLOS authors have the option to publish the peer review history of their article (what does this mean?). If published, this will include your full peer review and any attached files.

Reviewer #2: **Yes: **Sunil Sapkota

---

## [Author Response · Author response to Decision Letter 2]

4 Apr 2024

Dear Academic Editor of Plos One Journal

Thank you very much for your informative comments in our manuscript entitled:

Knowledge and perceptions of snakes, snakebites and their management among health workers in Sudan.

Hereby the responses for your comments:

1. The references were formatted correctly using the same style.

2. Four studies from other African countries were added.

3. The suggestions were in the attached manuscript file were included.

4. In Table 1, the average age (29.3 ± 8) deleted, and indicated in the main text.

5. The ages represented in some meaningful ranges. 

6. In table 3, the first aspect (Type of snakes) removed.

Corresponding author

Ali Awadallah Saeed

---

## [Editor Report · Decision Letter 3]

8 Apr 2024

Knowledge and perceptions of snakes, snakebites and their management among health workers in Sudan.

PONE-D-23-32936R3

Dear Dr. Saeed,

We’re pleased to inform you that your manuscript has been judged scientifically suitable for publication and will be formally accepted for publication once it meets all outstanding technical requirements.

Kind regards,

Timothy Omara, PhD

Academic Editor

PLOS ONE
---

## [Editor Report · Acceptance letter]

25 Jul 2024

PONE-D-23-32936R3 

PLOS ONE

Dear Dr. Saeed, 

I'm pleased to inform you that your manuscript has been deemed suitable for publication in PLOS ONE. Congratulations! Your manuscript is now being handed over to our production team.

Kind regards, 

on behalf of

Dr. Timothy Omara 

Academic Editor

PLOS ONE